# Stacked Bidirectional Convolutional LSTMs for 3D Non-contrast CT Reconstruction from Spatiotemporal 4D CT

**Sil C. van de Leemput, Mathias Prokop, Bram van Ginneken, Rashindra Manniesing**
Department of Radiology and Nuclear Medicine
Radboud University Medical Center
6525 GA Nijmegen, The Netherlands
`sil.vandeleemput@radboudumc.nl`

## Abstract

The imaging workup in acute stroke can be simplified by reconstructing the non-contrast CT (NCCT) from CT perfusion (CTP) images, resulting in reduced workup time and radiation dose. This work presents a stacked bidirectional convolutional LSTM (C-LSTM) network to predict 3D volumes from 4D spatiotemporal data. Several parameterizations of the C-LSTM network were trained on a set of 17 CTP-NCCT pairs to learn to reconstruct NCCT from CTP and were subsequently quantitatively evaluated on a separate cohort of 16 cases. The results show that C-LSTM network clearly outperforms basic reconstruction methods and provides a promising general deep learning approach for handling high-dimensional spatiotemporal medical data.

## 1  Introduction

CT is the preferred modality in a stroke imaging workup since fast diagnosis is critical for patient outcome. A stroke workup consists of a non-contrast CT (NCCT) scan to identify hemorrhages, is followed by a CT Angiography (CTA) to assess the cerebral vasculature, and is often followed by a CTP to differentiate core (irreversibly damaged brain tissue) and penumbra (salvageable tissue) [1]. The CTA is a 3D acquisition and the CTP is a 4D acquisition both after injection of contrast agent. A simplification of this stroke workup can be achieved by only acquiring a CTP and subsequently derive the NCCT and CTA from the CTP thereby reducing radiation dose, contrast usage and workup time. In principle this is feasible because the CTP contains more information than the CTA and NCCT. In previous work the feasibility of deriving high-quality CTA from CTP was shown [2].

In this work we present a novel convolutional LSTM (C-LSTM) neural network which is designed for 3D reconstruction from 4D spatiotemporal data. To validate the model it was applied to a patient dataset to reconstruct the NCCT from the CTP and compared to a baseline. The contributions of this work are twofold: We show the potential of C-LSTM for 3D reconstruction from 4D spatiotemporal data, and present the first application for NCCT from CTP reconstruction, which has the potential to simplify current stroke workup.

### 1.1  Related Work

C-LSTM is a type of recurrent neural network which combines the long short-term memory (LSTM) network [3] – the standard for processing sequential data – with convolution neural networks [4] – the standard for processing spatial data — by replacing the internal matrix multiplications of the weights with the input and hidden states with convolutional operations. This is different from methods

1st Conference on Medical Imaging with Deep Learning (MIDL 2018), Amsterdam, The Netherlands.

stacking normal LSTM networks on top of conventional convolutional layers, but these are often found under the same name in the literature. Hence, a single C-LSTM network is geared towards encoding both spatial and temporal features, including motion features, while preserving long-term recurrent dependencies.

The C-LSTM model has been first introduced in [5] to predict the weather from video sequences. The model has appeared in a variety of video analysis applications since then. Some works use C-LSTM to estimate human pose or gestures from video [6], [7]. For full sequence to sequence prediction from videos, some have integrated C-LSTM within auto encoders for abnormality detection [8] and next frame prediction [9]. Recurrent convolutional networks have been applied to generate super resolution video from low resolution video [10]. Despite the many interesting applications of the C-LSTM, it has not yet been applied to medical image reconstruction, nor has it been applied to CTP data. Furthermore, most C-LSTM applications have been limited to 3D spatiotemporal video data and were not designed to deal with 4D dynamic volumetric data.

Several CNNs for medical image reconstruction exist in the literature reporting overall small improved performance over traditional reconstruction approaches (see [11] for a review). The CNN methods typically employ a regression approach, i.e. optimizing the l2 loss between target and reconstruction. Nie et al. [12] reconstruct CT from MRI images using four 3D convolutional layers. Bahrami et al. [13] uses a CNN network for 7T from 3T MRI reconstruction with four 3D convolutional layers. Others use CNNs to perform low-dose reconstruction from CT [14], [15] on 2D CT images. However, the majority of the proposed regression approaches only cover 2D or 3D images and are not designed to account for the temporal information of the CTP.

## 2 Methods

### 2.1 C-LSTM

The convolutional LSTM model (C-LSTM) is an adaptation of the normal LSTM model and can be described by the following equations.

$$
\begin{aligned}
i_t &= \sigma(x_t *_x W_{xi} + h_{t-1} *_h W_{hi} + b_i) \\
f_t &= \sigma(x_t *_x W_{xf} + h_{t-1} *_h W_{hf} + b_f) \\
o_t &= \sigma(x_t *_x W_{xo} + h_{t-1} *_h W_{ho} + b_o) \\
g_t &= \phi(x_t *_x W_{xc} + h_{t-1} *_h W_{hc} + b_c) \\
c_t &= f_t \odot c_{t-1} + i_t \odot g_t \\
h_t &= o_t \odot \omega(c_t)
\end{aligned}
\tag{1}
$$

where, $x_t$ and $h_{t-1}$ are the inputs at time point $t$, with $x_t$ the input sequence data at time point $t$ and $h_{t-1}$ the previous hidden state. $h_t$ is the output at time point $t$ and also the hidden input state for the next time point $t + 1$. $i_t$, $f_t$, $o_t$, $g_t$ are respectively the input gate, the forget gate, the output gate, and the cell state, which encode how much the input at the current time point and the hidden state from previous time point contribute to the current cell state $c_t$, through the weight matrices $W_x$, $W_h$, and biases $b$. Usually, $\sigma$ and $\phi$ are respectively the sigmoid and hyperbolic tangent functions. $\odot$ is the element wise product and $*_x$ and $*_h$ are the convolution operators for the input and the recurrent input respectively. $\omega$ was set to the hyperbolic tangent function.

Note that the convolutional LSTM is essentially a generalization of the conventional LSTM and it can be obtained by setting a convolutional kernel of $1^3$ for $*_x$ and $*_h$ or by entirely replacing the convolutions with a matrix multiplication.

### 2.1.1 C-LSTM Layer

Since the C-LSTM falls within the class of recurrent neural networks it can have any of the following input output sequence mappings: one-to-one, one-to-many, many-to-one, many-to-many. However, in this work we only consider two variants which encapsulate the previous equations 1 in a single layer. This results in a function $F : S \rightarrow S$ which takes in a sequence $S$ of length $l$ and outputs an equally lengthy sequence $(h_1, h_2, \ldots, h_{l-1}, h_l)$ or alternatively only the last state in the sequence $h_l$.

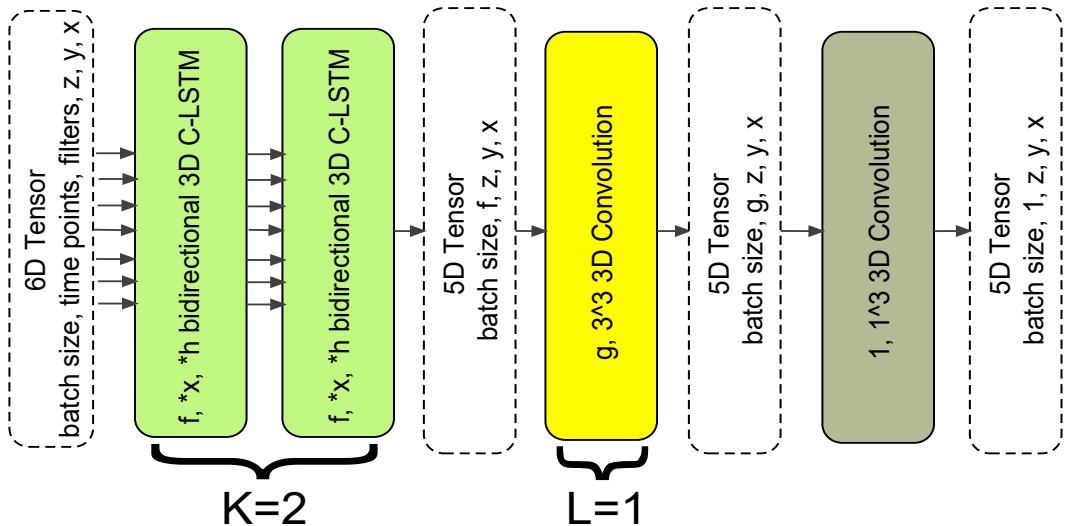

Figure 1: Parameterizable $K, f, *_x, *_h, L, g$- stacked C-LSTM network. In green $K$ bidirectional C-LSTM layers with $f$ filters, $*_x$ the size of the input kernel, and $*_h$ the size of the recurrent kernel. In yellow $L$ 3D convolution layers with $g$ filters, $3^3$ kernel and ReLU activation function. In gray a single 3D convolution with one filter, $1^3$ kernel and identity activation function. On the left the input sequence is a 6D Tensor, which is reduced to a 5D tensor with $f$ filters after the last C-LSTM layer. Subsequently is turned into a 5D tensor with $g$ filters after the $L$ convolution layers and the final output is a single filter 5D tensor.

### 2.1.2 Bidirectional C-LSTM

In a bidirectional approach to sequences, the signal is processed both from 0 to N and also from N to 0 by another similar recurrent network. Finally, the results are combined to generally yield better results. Since the entire sequence length is known beforehand in our case, we utilize the bidirectional approach and use summation the sequences at the end, i.e.: given sequence output 1 $(h_1^1, h_2^1, \ldots, h_{l-1}^1, h_l^1)$ and reversed but same sized sequence output 2 $(h_l^2, h_{l-1}^2, \ldots, h_2^2, h_1^2)$, the output of the combined bidirectional LSTM becomes: $(h_1^1 + h_l^2, h_2^1 + h_{l-1}^2, \ldots, h_{l-1}^1 + h_2^2, h_l^1 + h_1^2)$.

### 2.1.3 Stacked C-LSTM

The previously described components were combined in a reconstruction network consisting of a parameterizable stack of C-LSTM layers and convolutions. A schematic overview of the $K, f, *_x, *_h, L, g$- stacked C-LSTM network is shown in Figure 1.

The network takes a 3D spatiotemporal input sequence as a 6D tensor with as dimensions: batch size, time points, number of filters, and the spatial dimensions (z, y, x). The sequence is fed through a stack of $K$ bidirectional C-LSTM layers, each with $f$ filters, $*_x$ input convolution kernel size, and $*_h$ hidden convolution kernel size. All $K$ C-LSTM layers pass on the entire length of the sequence except for the last layer in the stack, which only passes on the last prediction, reducing the input to a 5D tensor with a filter size of $f$.

Next, the signal is fed into an optional stack of $L$ 3D convolutions with each $g$ filters, $3^3$ convolutional kernel, and a ReLU activation function. Finally, a last single 3D convolution with 1 filter, a $1^3$ convolutional kernel and an identity activation function is used to produce the output reconstruction as a 5D tensor.

### 2.2 Model Training

The training of the model employs a regression training scheme where the mean squared error loss between the NCCT and reconstructed NCCT was minimized using the RMSProp optimizer. The RMSProp optimizer was chosen since initial experiments yielded more stable training performance

than the SGD optimizer. The optimizer settings were kept at the default values. Each model was trained for 500 iterations, for which each iteration consists of 100 randomly sampled CTP sub-volumes from within the cranial cavity mask over all training set cases. Training was performed on an NVIDIA Titan X GPU with 12 GB of RAM using Theano [16] as backend.

The initial hidden state of each C-LSTM layer was set to all zeros. The weights $W_{xi}, W_{xf}, W_{xo}, W_{xc}$ where all initialized using uniform Xavier initialization [17]. The recurrent kernels $W_{hi}, W_{hf}, W_{ho}, W_{hc}$ were all initialized using random othogonal matrices. All bias terms were set to 0, except for the forget bias, which was set to 1 as recommended by [18]. All normal convolutional layers were initialized using uniform Xavier initialization.

### 2.3 Implementation Details

The stacked 3D C-LSTM models have been implemented in Keras [19]. The C-LSTM operations at each time point have been optimized by exploiting that $i_t, f_t, o_t$ and $g_t$ from equation 1 require similar computations. Hence, the convolution operations $*_h, *_x$ can be computed efficiently by concatenating the weight matrices for $W_x$ and $W_h$, i.e.: $x_t *_x \{W_{xi}, W_{xf}, W_{xo}, W_{xc}\}$ and $h_{t-1} *_h$ $\{W_{hi}, W_{hf}, W_{ho}, W_{hc}\}$. In this way the components for $i_t, f_t, o_t, g_t$ can be computed by just two convolutions instead of eight.

## 3 Data

This retrospective study included 39 patients (age $66 \pm 13$ years, $64\%$ male) with suspicion of stroke admitted to our hospital in 2015 and 2016 and who have received both a NCCT and a CTP scan. Six cases had major pathology like bleeds and large infarcts. The data was split into a training set of 17 cases, validation set of 6 cases, and a test set of 16 cases.

CTPs were acquired on a 320-row CT scanner (Toshiba Aquilion ONE, Japan) consisting of 19 volumetric scans with different exposures per time point. Patients received 80 mL of contrast agent (Iomeron) injected in the cephalic vein at the start of the first acquisition. Image reconstruction was done using a FC41 smooth convolution kernel, resulting in $512 \times 512 \times 320$ voxels with a voxel size of $0.47 \times 0.47 \times 0.5$ mm. NCCTs were acquired on the same scanner reconstructed with a FC26 kernel yielding $512 \times 512 \times 302$ voxels with a voxel size of $0.43 \times 0.43 \times 0.5$ mm.

### 3.1 Preprocessing

All CTP time points $t > 0$ were rigidly registered to the first CTP time point ($t = 0$), to correct for potential head movement during acquisition. The registration was performed using the method and parameter settings as described by [20]. The NCCT was rigidly registered to the same space of the first time point of the CTP with Elastix [21] using similar settings. A cranial cavity mask was created using the method of [22] to segment all intracranial soft tissue. The final cranial cavity mask was obtained by discarding all voxels with an intensity below air density ($-1000$ HU) followed by a binary erosion with a 3D ball structuring element with a radius of three voxels. Finally, before neural network training and prediction the input was linearly mapped from $[-50, 200]$ to $[0, 1]$ and the mapping was reversed after training and prediction.

## 4 Evaluation

### 4.1 Quantitative evaluation

All methods were compared using the following regression error measures: mean absolute error (MAE), mean squared error (MSE), explained variance score (EV), and $r^2$ score. However, the diagnostic relevant information of a NCCT are only found within the cranial cavity. Hence, only the voxels within the cranial cavity mask (described in section 3.1) were used for computing these quantitative metrics.

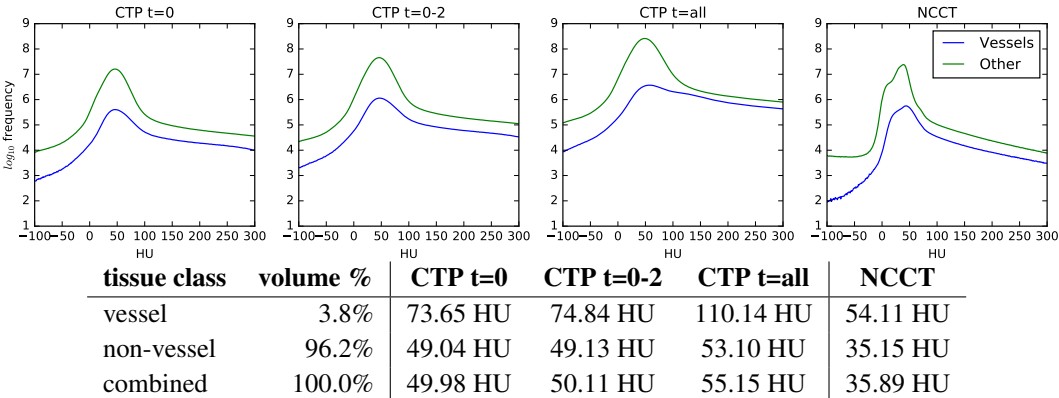

| tissue class | volume % | CTP t=0 | CTP t=0-2 | CTP t=all | NCCT |
|---|---|---|---|---|---|
| vessel | 3.8% | 73.65 HU | 74.84 HU | 110.14 HU | 54.11 HU |
| non-vessel | 96.2% | 49.04 HU | 49.13 HU | 53.10 HU | 35.15 HU |
| combined | 100.0% | 49.98 HU | 50.11 HU | 55.15 HU | 35.89 HU |

Figure 2: Intensity histograms of Hounsfield unit (HU) within the $[-100, 300]$ domain averaged over the entire dataset of 39 patients. The dataset is examined for different time point partitions of the CTP (t=0, t=0-2, and t=all) and the NCCT for vessel and non-vessel tissues within the cranial cavity. The table shows the average HU for the CTP partitions and NCCT images.

## 4.2 Baseline Reconstruction Models

A lower bound baseline was established using three naive reconstruction methods from the CTP: taking the first time point (t=0), taking the mean of the first 3 time points (t=0-2), and taking the average over all time points (t=all). Taking the first time point is an obvious approach for the reconstruction, since it is the CTP time point with the highest exposure and hence has is the time point with the best signal to noise ratio. Also earlier in the time sequence the contrast agent is less expressed, which is closer to the signal intensity of the target NCCT.

An analysis of the cranial cavity intensities histograms over all patients within the domain of $[-100, 300]$ HU on the CTP and NCCT shows a basic intensity bias between the two imaging types. The intensity histograms can be found in Figure 2 and are divided in vessel and non-vessel tissue type counts. The voxels belonging to the vessel class were determined by assigning all voxels with values > 110 of the temporal variance of the CTP.

This bias must be accounted for when computing the quantitative evaluation metrics, to avoid underestimating the reconstruction quality of these models. The bias was estimated as the difference in average non-vessel tissue intensity of the NCCT and respective CTP class, which were $-13.9$, $-14.0$, and $-18.0$ HU for respectively t=0, t=0-3 and t=all. The learned models in this work do not suffer from this bias, since these methods learn to estimate this bias from the data.

## 5 Experiments

Four differently parameterized stacked C-LSTM architectures were trained for $500$ iterations. The different parameterizations of the stacked C-LSTM architecture are shown in Table 1.

## 6 Results

Table 1 and Figure 3 show the results on the test set of the four experiments with the parameterized stacked C-LSTM models after completing training. The trained stacked C-LSTM models (exp. 1-4) significantly outperformed the three baseline methods (t=0, t=0-2, t=all) with $p < 0.01$ (paired samples t-test) on all performance metrics. Between the four different parameterizations experiment 2 outperformed all the other methods on all the metrics, followed by experiment 4, then experiment 1, and finally experiment 3. Paired samples t-tests showed significant differences $p < 0.01$ on all reconstruction performance metrics between all parameterizations, except for experiment 1 and 3, where the MSE and r$^2$ score had a $p > 0.05$.

Table 1: Experiments with stacked C-LSTM parameter settings, spatial input size and batch size used for training, and average test set performance after $500$ iteration on mean absolute error (MAE), mean squared error (MSE), explained variance (EV), and $r^2$ score. The baseline reconstruction models (t=0, t=0-2. t=all) have been added for reference.

| exp. | input | batch | K | $*_x$ | $*_h$ | f | L | g | MAE | MSE | EV | $r^2$ |
|------|-------|-------|---|-------|-------|---|---|---|------|-------|--------|--------|
| 1 | $40^3$ | 2 | 1 | $3^3$ | $1^3$ | 64 | 0 | - | 7.17 | 96.36 | 0.530 | 0.470 |
| 2 | $40^3$ | 2 | 1 | $3^3$ | $3^3$ | 64 | 0 | - | 6.17 | 71.96 | 0.625 | 0.606 |
| 3 | $30^3$ | 5 | 2 | $3^3$ | $1^3$ | 40 | 1 | 50 | 7.48 | 101.32 | 0.547 | 0.444 |
| 4 | $30^3$ | 5 | 3 | $3^3$ | $1^3$ | 40 | 1 | 50 | 6.53 | 81.60 | 0.581 | 0.552 |
| t=0 | - | - | - | - | - | - | - | - | 10.28 | 194.03 | -0.013 | -0.076 |
| t=0-2 | - | - | - | - | - | - | - | - | 9.77 | 177.40 | 0.078 | 0.014 |
| t=all | - | - | - | - | - | - | - | - | 10.64 | 288.68 | -0.577 | -0.626 |

Figure 4 shows some qualitative slices from the test set of the temporally averaged input CTP volumes, target NCCT volumes, reconstructed NCCT volumes by the stacked C-LSTM model from experiment 2, and the difference volume between target and reconstruction.

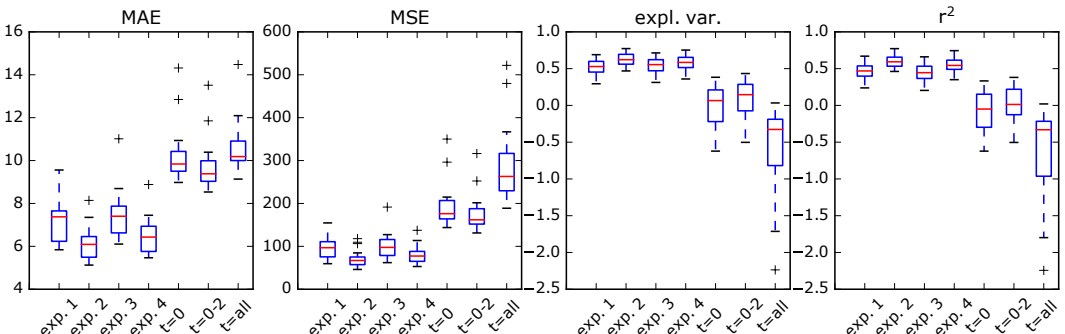

Figure 3: Final results on the test set (16 cases) for all methods: experiment 1-4 (exp. 1-4) and the 3 baseline methods (t=0, t=0-2, t=all). Metrics from left to right: mean absolute error (MAE), mean squared error (MSE), explained variance (EV), and r$^2$ score (r$^2$).

## 7  Discussion

In this work we have presented a stacked C-LSTM architecture for 3D reconstruction from 4D spatiotemporal data. Furthermore, we have shown that using four different parameterizations of this model we were able to reconstruct the NCCT from the 4D spatiotemporal CTP data with better performance on MAE, MSE, explained variance score, and r$^2$ score than three baseline methods (Figure 3).

Figure 4 gives a good impression of the expressiveness of the C-LSTM model to encode both the spatial and the temporal information. In general the vessels were completely suppressed in the final model predictions, but the calcification traces (bright small spots on the NCCT and reconstruction seen in the top row image), were not. When contrasting the temporal average of the CTP with the reconstruction it can be seen that the model was able to overcome the general intensity bias from the CTP with respect to the NCCT target. The model also creates better contrast of the cerebrospinal fluid at the giri and sulci with the brain tissue. Furthermore, the reconstruction contains much less noise and produces a smoother result, which might be relevant for finding diagnostic markers.

Figure 3 shows the expected results between the baseline methods, where t=0-2 performed best followed by t=0 and t=all performed the worst. The earlier time points (t=0, t=0-2) show less expression of the contrast agent – because the contrast agent is injected approximately around the first time point t=0 and it takes some time to circulate – and generally have a better signal to noise ratio, but a single time point (t=0) contains more noise than averaging over multiple time points (t=0-2) at the start of the CTP sequence.

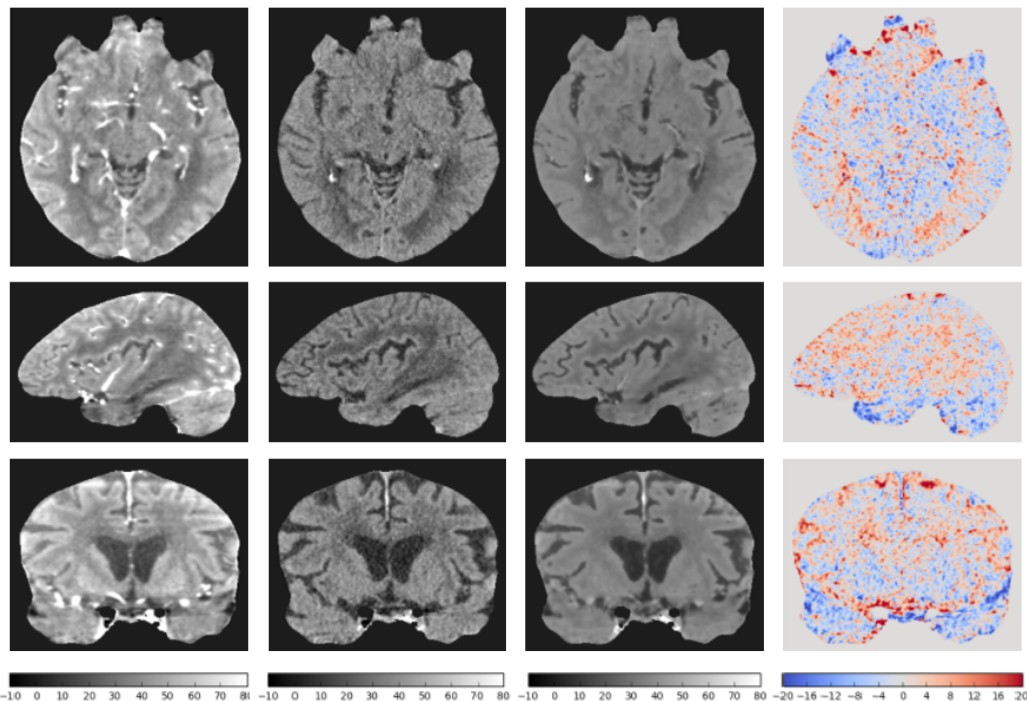

Figure 4: Qualitative results on three different cases from the test set showing slices of the intracranial tissue. From top to bottom: an axial, coronal, and sagittal slice. From left to right: Temporal average of CTP, target NCCT, reconstructed NCCT by best trained model from experiment 2, and difference image between target and reconstruction. The scales at the bottom are in Hounsfield units (HU).

Table 1 shows that the best performing parameterizations (experiment 2) of the stacked C-LSTM model was just a single C-LSTM layer with a recurrent kernel size of $3^3$ ($*_h = 3^3$), which improved performance over experiment 1, which had the same settings except a smaller recurrent kernel size of $1^3$. This finding is somewhat surprising, since the CTP data do not necessarily show much motion between time points, which justifies the smaller recurrent kernel size better. However, upon closer examination, the bigger kernel size might compensate for minor intra registration errors of the CTP time points with t > 0 to the first time point t=0. Furthermore, comparing experiment 3 ($K = 2$) and experiment 4 ($K = 3$) suggests that stacking more C-LSTM layers helps to achieve better performance. The parameterization and extension of the stacked C-LSTM architecture is still open for much experimentation, but this remains for future work.

The C-LSTM is better suited for spatiotemporal data than CNN and LSTM methods separately. While it is possible to parse sequential data using CNNs [23], it is not a natural fit and requires some workarounds. Also, parsing spatiotemporal data with only LSTM using flattened spatial data would make it more difficult to encode spatial features like edges.

The proposed C-LSTM models and training scheme are not limited to the application of NCCT reconstruction and could be utilized for other applications involving spatiotemporal data. This work employs a regression scheme for training, but it is easy to make it into a segmentation scheme, by adding a softmax to the model and changing the loss. The results show that the C-LSTM model was able to suppress the vessels within the CTP, but it might also be used to filter other information as well. Another interesting future direction is the use of the model for computing perfusion images.

To further the acceptance of the method as a replacement for a normal NCCT scan the evaluation of the reconstruction results could be extended with a qualitative assessment of diagnostic relevant information like: hemorrhages, dense vessel sign, and infarcts. This information could be graded for both the NCCT and the reconstructions by experienced radiologists and be compared to assess whether all diagnostically relevant information is still present in the reconstruction.

An interesting direction is to integrate the C-LSTM network within a generative adversarial network (GAN) [24] for reconstruction [25]–[27]. In this setting, two networks are trained in competition, a generator which tries to generate images looking similar to a target image distribution and a discriminator which tries to distinguish between images made by the generator and the real images. A well trained generator can create images looking very close to the real target data distribution intrinsics. However, while a generator might be better able to mimic the target data distribution this is not necessarily the signal of interest. For example, if the target image contains a lot of noise, the generator will start to mimic this to be able fool the discriminator and give high error on normal regression error measures (like the Euclidean distance) and induces noise which suppresses potential important diagnostic findings. This work does not employ a GAN scheme, since the NCCT is inherently very noisy and a noise free signal to clearly distinguish diagnostic findings is desirable.

To the best of our knowledge we have presented the first deep learning application of C-LSTM for 3D NCCT reconstruction from 4D spatiotemporal CTP, which could potentially simplify current stroke workup. Furthermore, the proposed C-LSTM models and training scheme pose promising tools for handling spatiotemporal data in medical imaging and can be used for other problems as well.

### Acknowledgments

This work was supported by research grants from the Netherlands Organization for Scientific Research (NWO), the Netherlands and Canon Medical Systems Corporation, Japan.

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
