# OpenReview forum: "Stacked Bidirectional Convolutional LSTMs for 3D Non-contrast CT Reconstruction from Spatiotemporal 4D CT"
_MIDL.amsterdam/2018/Conference — MIDL 2018 Oral_

### Review · AnonReviewer1 · 2018-05-08
**C LSTM for modality synthesis**

**Rating:** 4
**Confidence:** 2

**Review:**

The author aim at extracting non-contrast CT (and potentially CT angio) from CT perfusion scans. They follow a generalized LSTM approach to track the changes resulting from the tracer dynamics. Good validation.

Pro: Innovative application for 'modality synthesis'. Successfully illustrates 4D learning of CT tracer dynamics which can be considered a hard learning problem. Use of C LSTM is novel.

Con: In theory, one can set up perfusion scans that contain native CT and CTA naturally as part of their 4D sequence - although at a very high radiation dose for the patient.  To this end perfusion often has low spatial or temporal resolution - or even field of view - in the clinical setting, and CTA and native CT do offer complementary information that is not contained (at that resolution, contrast, or detail) in the clinical perfusion data.

**Special Issue:**

Yes

---

### Review · AnonReviewer2 · 2018-05-09
**This paper presents the reconstruction method of 3D non-contrast CT from 4D CT by using stacked bidirectional convolutional LSTMs. Although authors have experimented on the small scale dataset, the problem-solving approach is novel and the experimental results are interesting.**

**Rating:** 3
**Confidence:** 1

**Review:**


Quality & Clarity

#1. This paper is well organized.
#2. Experimental results are well organized and interesting.
#3. The detailed analysis have been described.

Originality & Significance

(+) Although it is an existing method, it have been applied to newly defined problems.
(-) Authors have evaluated the proposed algorithm with small scale test set.

**Special Issue:**

No

---

### Comment · ~Bram_van_Ginneken1 · 2018-05-18
**Selection for longlist for special issue Medical Image Analysis**

Dear authors,

Congratulations on your acceptance to MIDL! We have selected your paper on the longlist for the Medical Image Analysis Special Issue. Please read this page:
https://midl.amsterdam/special-issue-in-medical-image-analysis/
Please answer the three questions that are listed on that page about your interest in submitting to the special issue, potential overlap with other publications, and related publications.

You can post your answer here directly below on openreview.net, or mail me directly at bram.vanginneken@radboudumc.nl.

Best regards, Bram

---

### Decision · Program_Chairs · 2018-05-15
**Paper107 Acceptance Decision**

Oral